# Profiling of Steroid Metabolic Pathways in Human Plasma by GC-MS/MS Combined with Microwave-Assisted Derivatization for Diagnosis of Gastric Disorders

**DOI:** 10.3390/ijms22041872

**Published:** 2021-02-13

**Authors:** Wonwoong Lee, Hyunjung Lee, You Lee Kim, Yong Chan Lee, Bong Chul Chung, Jongki Hong

**Affiliations:** 1Department of Pharmacy, College of Pharmacy, Woosuk University, Wanju 55338, Korea; wwlee@woosuk.ac.kr; 2Department of Pharmacy, College of Pharmacy, Kyung Hee University, Seoul 02447, Korea; e2_hj@ktr.or.kr (H.L.); uiriuiri@naver.com (Y.L.K.); 3Department of Internal Medicine, Institute of Gastroenterology, Yonsei University College of Medicine, Seoul 03722, Korea; LEEYC@yuhs.ac; 4Molecular Recognition Research Center, Korea Institute of Science and Technology, Seoul 02792, Korea

**Keywords:** endogenous steroids, human plasma, solid-phase extraction, microwave-assisted derivatization, GC-MS/MS-dMRM, profiling analysis, gastric cancer

## Abstract

Steroid hormones are associated in depth to cellular signaling, inflammatory immune responses, and reproductive functions, and their metabolism alterations incur various diseases. In particular, quantitative profiling of steroids in plasma of patients with gastric cancer can provide a vast information to understand development of gastric cancer, since both sex hormones and glucocorticoids might be correlated with the pathological mechanisms of gastric cancer. Here, we developed a gas chromatography-tandem mass spectrometry-dynamic multiple reaction monitoring (GC-MS/MS-dMRM) method combined with solid-phase extraction (SPE) and microwave-assisted derivatization (MAD) to determine 20 endogenous steroids in human plasma. In this study, MAD conditions were optimized with respect to irradiation power and time. The SPE enabled effective cleanup and extraction for profiling of steroid hormones in human plasma samples. The MAD could improve laborious and time-consuming derivatization procedure, since dielectric heating using microwave directly increase molecular energy of reactants by penetrating through medium. Furthermore, dMRM method provided more sensitive determination of 20 steroids, compared to traditional MRM detection. The limits of quantification of steroids were below 1.125 ng/mL and determination coefficients of calibration curves were higher than 0.9925. Overall precision and accuracy results were below 19.93% and within ±17.04%, respectively. The developed method provided sufficient detection sensitivities and reliable quantification results. The established method was successfully applied to profile steroid metabolism pathways in plasma of patients with chronic superficial gastritis (CSG), intestinal metaplasia (IM), and gastric cancer. Statistical significances of steroid plasma levels between gastric disorder groups were investigated. In conclusion, this method provided comprehensive profiling of 20 steroids in human plasma samples and will be helpful to discover potential biomarkers for the development of gastric cancer and to further understand metabolic syndrome.

## 1. Introduction

Steroid hormones are generally biosynthesized from cholesterol in the adrenal glands, gonads, brain, placenta, and adipose tissues, and are circulated via bloodstream in the human body [1]. It is well known that steroids are deeply involved in most physiological activities such as cellular signaling, sexual activity, reproductive functions, immune responses, and homeostasis [2,3]. The metabolism cascades of steroids are mainly regulated by two major enzymes (i.e., cytochrome P450 (CYP) and hydroxysteroid dehydrogenase (HSD)) (Figure 1) [4]. Various physiological diseases including congenital adrenal hyperplasia, hypertension, and cancer can be correlated with dysregulation of these enzymes, which lead to alterations of steroid metabolism pathways. In particular, several enzymes (such as CYP19A1, HSD3B1, and steroid sulfotransferase 2B1b] are involved in progression of gastric cancer [5,6,7]. In addition, since gastric cancer has been predominantly prevalent in male patients, it is speculated that sex steroid hormones play a crucial role in development of gastric cancer [8]. Nevertheless, it is still unclear pathological relation between gastric cancer and steroid hormones. Therefore, quantitative profiling of steroids in biological samples can provide a vast information of uncertain pathological mechanisms of gastric cancer.

Since mass spectrometry (MS) could provide high detection selectivity and sensitivity, analytical methods based on MS combined with chromatographic separations have been widely employed to determine endogenous steroid metabolites in biological samples [9,10,11]. Although liquid chromatography (LC)-MS approaches have been extensively used due to applicability of simple and easy sample preparation, gas chromatography (GC)-MS methods could provide better chromatographic resolution and ionization reproducibility. Hence, the GC-MS methods are considered as “gold standard” in steroid profiling [12,13]. Moreover, dynamic multiple reaction monitoring (dMRM) method using tandem MS can improve identification and quantification of trace level metabolites in biological samples, due to selective detection of target analytes within retention time windows.

Nonetheless, GC-MS approaches should be employed along with fine sample preparation including extraction, purification, and chemical derivatization, which is laborious and time-consuming. For instance, protein precipitation, liquid-liquid extraction, solid-phase extraction (SPE), and supported liquid extraction (SLE) methods have been performed to extract steroid hormones from biological samples prior to GC-MS analysis [14,15,16,17,18,19]. Among them, SPE packed with C18 sorbents could provide effective cleanup, extraction, and even concentration for profiling of endogenous steroids in biological samples. In addition, chemical derivatization should be performed to improve thermal stability, volatility, and chromatographic performance of steroid hormones [4]. To effectively derivatize steroids, several derivatization methods were compared in terms of derivatization reagents, reaction time, and heating conditions (i.e., conventional block heating, sonication, and microwave) [20,21]. Among heating techniques, microwave-assisted derivatization (MAD) was proved as the most efficient derivatization method [21], since microwave can directly increase reaction energy of analytes and derivatization reagents by dielectric heating.

According to previous studies, steroid concentration levels in biological samples have provided insights to better understand pathophysiological mechanisms in various diseases. For instance, urinary steroid signatures using hierarchically clustered heat map analysis could be used for diagnosis of benign prostatic hyperplasia [22]. In addition, by urinary steroid profiling, it was reported that increased androgen and glucocorticoid levels might be a characteristic marker for adrenocortical malignant [23]. Interestingly, since the 17β-estradiol/estrone ratio in cancerous breast tissue was substantially higher than the ratio in benign breast tissues, sex steroid hormone measurement in breast tissue can be used as assessment indicator [24]. Additionally, steroid plasma levels in cancer patients can be utilized to evaluate anticancer drug efficacy [25,26]. Therefore, steroid profiling results of biological samples will be helpful to find potential biomarkers for various diseases [27].

Herein, we developed a GC-MS/MS-dMRM method combined with SPE and MAD to profile 20 endogenous steroids in human plasma. The developed analytical method was applied to 66 plasma samples of patients with chronic superficial gastritis (CSG, *n* = 20), intestinal metaplasia (IM, *n* = 13), and gastric cancer (*n* = 33) to find potential biomarkers for gastric cancer. Obtained quantified values of gastric disorder groups were analyzed by significance analysis. This method provides comprehensive profiling of 20 endogenous steroids in human plasma and will be a promising tool to discover diagnostic biomarkers of gastric cancer. Furthermore, this study is expected to provide insights to understand pathological mechanisms of gastric cancer and other steroidal metabolic disorders.

## 2. Results and Discussion

### 2.1. SPE and Microwave-Assisted Derivatization (MAD) Procedure

To extract endogenous steroids and reduce interference materials from complex biological samples, an SPE method was widely employed as an effective sample cleanup method [28,29,30]. Therefore, in this study, the SPE cartridge packed with C18 sorbents was used to extract 20 steroids in human plasma samples. The SPE protocol was performed according to a manufacturer’s instruction and a previously developed method [28] with slight modification. In conclusion, overall recoveries of endogenous steroids in plasma samples were shown similarly with previous reports (87–101%).

For sensitive analysis of steroids in biological samples, it is essential to employ derivatization of steroids to obtain better volatility, peak separation, and detection sensitivity. On chemical structures of steroids, ketone groups do not have to be derivatized to introduce a GC-MS, while hydroxyl groups should be protected to improve volatility and thermal stability [31]. For this reason, silylation reactions, such as trimethylsilylation (TMS), have been predominantly employed to derivatize hydroxyl groups of steroids. In preliminary test, several representative reagents for derivatization of hydroxyls were evaluated by derivatization yields (Figure 2). Among them, since MSTFA:NH_4_I:DTE (500:4:2, *v/v/v*) mixture provided high yields of steroid derivatives, the reagent mixture was chosen for further optimization of derivatization conditions.

Similarly, derivatization protocols have been developed to obtain better sensitivities and thermal stabilities of endogenous steroids. For example, microwave-assisted derivatization (MAD), sonication-assisted derivatization (SAD), and traditional heating block derivatization were assessed by comparing derivatization yields [21]. A MAD method provided increased relative response factor and reduced reaction time compared to thermal block heating and SAD, due to the use of dielectric heating. Moreover, for SAD, although heating using water bath increased response values compared to that using thermal block, no difference was observed between water bath heating with/without sonication. For this reason, we utilized MAD to derivatize targeted 20 endogenous steroids in human plasma.

A MAD reaction through dielectric heating could easily convert molecule into its activated state via microwave penetration [32]. In commercial microwave ovens using 2.45 GHz, the temperature increase of reactants is mainly influenced by microwave power and irradiation time. Therefore, a MAD protocol was optimized with respect to microwave power and irradiation time. To investigate optimized MAD conditions, six representative analytes (DHEAS, AN, adione, E2, PROG, and F) of 20 endogenous steroids were selected according to structural characteristics. Microwave power and irradiation time were evaluated by comparing relative response values in the range of 100–600 W and 1–5 min, respectively. As shown in Figure 3, six steroid derivatives were shown the highest relative response values, when microwave power and irradiation time were 300 W and 1 min, respectively. According to increase of irradiation time, relative response values of analytes were decreased. In addition, microwave power around 300 W exhibited similar results. Moreover, under optimized MAD conditions, a MAD method was compared with traditional heating method using thermal block. Six representative steroid derivatives showed a nearly identical relative response when the MAD method was compared with thermal block heating for 20 min (Appendix A). Consequently, the MAD method was efficient nearly 20 times, compared to traditional heating methods using thermal block.

### 2.2. GC-MS/MS-dMRM

Generally, an MRM method has been widely used to determine trace level of endogenous metabolites in biological samples. Using abundant and specific precursor-product ion transitions, MRM mode enables selective and sensitive determination of endogenous steroids in complicated biological samples without any serious effects of matrix contaminants [33]. Nevertheless, traditional MRM method has been suffered from its relatively low metabolite coverage and throughput capacity [34]. Via scanning every transition near the expected retention time of chromatographic run, scheduled strategy of MRM (namely dynamic MRM) could provide further increase of detection sensitivity and throughput [35]. Therefore, in this study, the dMRM method using GC-MS/MS was utilized to sensitively detect endogenous steroids in human plasma samples.

Through investigation of EI-MS spectra and MS/MS spectra for steroid derivatives, MRM transitions based on sensitive and reproducible MS/MS fragment ions were selected and further collision energy was optimized (Appendix A). Furthermore, to effectively detect MRM transitions of target steroids, dMRM method was employed. Since dMRM methods can selectively detect target MRM transitions only near their retention time windows, dMRM provided better detection sensitivity for endogenous steroids compared to traditional MRM. As shown in Figure 4, detection sensitivity of most steroids was increased almost 10 times, when MRM chromatograms using dMRM mode were compared with that using traditional MRM. In conclusion, the dMRM method provided better detection results compared to traditional MRM method.

### 2.3. Analysis of Human Plasma with CSG, IM, and Gastric Cancer

Using the developed method, 20 endogenous steroids in 66 human plasma samples were determined. For reliable quantification and confirmation results, MRM chromatograms of all target steroids and isotope labeled ISs were reconstructed, as shown in Figure 5. Individual MRM chromatogram for all analytes showed that 20 target steroids were sensitively detected without any severe interferences. Peak area ratios of endogenous steroids to corresponding ISs were investigated to quantify 20 steroids in human plasma and used to find metabolic alterations in steroid metabolic pathways. Most endogenous steroids could be quantified using this method. Calculated concentration levels and the *p*-values of target steroids in human plasma were summarized in Table 1. The *p*-values of target steroid plasma levels between groups were analyzed using R 4.0.3 (R Core Team, Vienna, Austria).

Using steroid concentration levels in plasma samples, we investigated to find appropriate biomarkers determining state of gastric disorders (such as CSG, IM, and gastric cancer). As shown in Figure 6A, based on concentrations of target steroids from plasma samples, E2 levels in plasma samples showed a significant difference between three gastric disorder groups. Among these groups, a statistical difference of E2 levels between IM and gastric cancer was confirmed via post hoc test. According to previous studies, it was reported that E2 inhibits gastric disorders including gastric cancer and *Helicobacter pylori*-induced pathology [36,37,38]. In this study, E2 plasma levels of IM group were suppressed compared to CSG and gastric cancer groups. We speculated that patients with IM (known as earlier stage of gastric cancer) are lost or losing their controlling capability of E2 against gastric cancer. Meanwhile, interestingly, E2 concentrations of gastric cancer patients were significantly higher than that of IM patients. This result carefully suggested that increased E2 plasma levels of gastric cancer patients might be associated with response against the gastric cancer tissues.

Moreover, the step-by-step metabolism state in steroid metabolic pathways were also investigated by analyzing concentration ratios of products to precursors, since the precursor-product metabolism state can reflect activity and/or expressions of metabolism enzymes relating to steroid metabolism [39]. In this study, results for significance analysis of precursor-product ratios were summarized in Table 2. As shown in Figure 6B,C, metabolism pathways related to CYP11B1 and HSD17B were significantly different between gastric disorder groups. Interestingly, HSD17B might be also associated with gastric cancer, since it was reported that HSD17B could be related to cancerous gastric mucosa [40,41]. Nevertheless, other steroid metabolism pathways related to CYP11B1 and HSD17B (such as F/17α-OH-PROG and A-diol/DHEA) were not shown significant differences between gastric disorder groups. In-depth studies on other multiple metabolism states and related enzymatic activity should be employed, since CYP11B1 and HSD17B have also catalytic activity in several steroid metabolism pathways.

Nonetheless, this method could provide reliable quantification results of 20 endogenous steroid hormones in human plasma samples. Moreover, this study demonstrated that gastric cancer might be sensitive to estrogen and associated with estrogen-related enzymes. In conclusion, this study suggested analytical method to profile endogenous steroids in human plasma and to find potential biomarkers for gastric disorders.

## 3. Materials and Methods

### 3.1. Chemicals and Materials

Most of authentic standards (21-deoxycortisol (21-DF), androstenedione (A-dione), estrone (E1), 17β-estradiol (E2), progesterone (PROG), dihydrotestosterone (DHT), cholesterol (CHOL), cortisol (F), cortisone (E), androsterone (AN), estriol (E3), 17α-hydroxypregnenolone (17α-OH-PREG), and 17α-hydroxyprogesterone (17α-OH-PROG)) and chemicals (ammonium iodide (NH_4_I), dithioerythritol (DTE), pyridine, trifluoroacetic acid (TFA), and methyl tert-butyl ether (MTBE)) were purchased from Sigma-Aldrich (St. Louis, MO, USA). Dehydroepiandrosterone sulfate (DHEAS), dehydroepiandrosterone (DHEA), and pregnenolone (PREG) were obtained from Avanti Polar Lipids (Alabaster, AL, USA). Androstenediol (A-diol), 7α-hydroxycholesterol (7α-OH-CHOL), and 7α-hydroxy DHEA (7α-OH-DHEA) were supplied from Steraloids (Newport, RI, USA). Testosterone (T) and dichloromethane were purchased from Junsei (Tokyo, Japan). Five isotopically labeled authentic steroids (testosterone-2,2,4,6,6-d_5_, cholesterol-2,2,3,4,4,6-d_6_, 17β-estradiol-2,4,16,16,17-d_5_, cortisol-9,11,12,12-d_4_, and progesterone-2,2,4,6,6,17α, 21,21,21-d_9_) were obtained from C/D/N Isotopes (Pointe-Claire, QC, Canada). As the trimethylsilyl (TMS) derivatization reagents, *N*-methyl-*N*-trifluorotrimethylsilylacetamide (MSTFA) and *N*,*O*-bis(trimethylsilyl)trifluoroacetamide (BSTFA) with 1% trimethylchlorosilane (TMCS) were purchased from Macherey-Nagel (Duren, Germany) and Supelco Chemical (Bellefonte, PA, USA), respectively. All organic solvents for sample preparation reagents such as ethyl acetate (EA), acetonitrile (ACN), and methanol (MeOH) were supplied from J.T. Baker (Rockford, IL, USA). De-ionized water (DW) was produced using Millipore Direct-Q3 purification system from the Millipore Corporation (Billerica, MA, USA). All chemicals used in this study were analytical grade or HPLC grade.

### 3.2. Preparation of Human Plasma

A total of 66 patients with different stages of gastric diseases (chronic superficial gastritis (CSG, *n* = 20), intestinal metaplasia (IM, *n* = 13), and gastric cancer (*n* = 33)) were enrolled in this study. All blood samples were collected through routine clinical test and categorized according to their gastric diseases. Collected human plasma samples were immediately frozen in liquid nitrogen and stored in deep freezer at –80 °C until analysis. All of sample information is summarized in Appendix A. All experimental procedures in this study were approved by the ethical committee of Severance Hospital, Seoul, Korea (4-2013-0880).

### 3.3. Solid-Phase Extraction (SPE) Procedure

Overall solid-phase extraction procedures were performed according to manufacturer’s instruction with slight modification. Two hundred microliter of collected plasma were transferred into 2 mL centrifuge tube, followed by addition of 5 µL of internal standard (IS) solutions (1 µg/mL). Then, plasma samples were diluted with 800 µL of 0.2 M acetate buffer at pH 5.2. SPE cartridges filled with C18 sorbent (Agilent, Santa Clara, CA, USA), which were sequentially pre-conditioned with 1 mL of MeOH and 1 mL of DW. After sample loading, the SPE cartridges were washed with 1 mL of DW, followed by dryness under vacuum (–30 kPa). Finally, all target steroids were eluted from dried SPE sorbents using 1 mL of EA.

### 3.4. Microwave-Assisted Derivatization (MAD) Procedure

Sample solution eluted from SPE cartridges was dried under gentle stream of nitrogen gas to remove trace water. The silylation reagents mixture (MSTFA:NH_4_I:DTE, 500:4:2, *v/v/v*) was added into dried sample vials. Then, sample vials were transferred to commercial microwave oven (Samsung Electronics, Seoul, Korea), which is able to manually control irradiation power and time. For silylation of 20 target steroids, derivatization reactions were performed under optimized irradiation power and time: 300 W and 60 s. After microwave-assisted derivatization (MAD), sample vials were transferred to auto-sampler and sample aliquots (2 µL) were injected into GC-MS/MS.

### 3.5. GC-MS/MS-MRM Conditions

GC-MS/MS analyses were employed by an Agilent 7890B gas chromatograph combined with an Agilent 7000C triple quadrupole mass spectrometer (Agilent, Palo Alto, CA, USA). Sample aliquot (2 μL) was injected into the injection port heated at 280 °C with split mode (5:1). An HP-ULTRA1 capillary column (17 m × 0.2 mm i.d., 0.11 µm film thickness) was installed to separate 20 target steroids. As a carrier gas, helium gas (99.999%) was set at a flow rate of 0.8 mL/min. The oven temperature was initially 150 °C, ramped to 230 °C at 20 °C/min and held for 2 min, increased to 250 °C at 2 °C/min and held for 2 min, and then finally elevated to 310 °C at 30 °C/min and held for 2 min. For dynamic multiple reaction monitoring (dMRM) detection, characteristic ions and retention times of 20 steroids, as silylated derivatives, were investigated with full scan mode (*m/z* 70–750). The mass spectrometric conditions were set as follows: electron ionization (EI) energy at 70 eV, ion source temperature at 230 °C, and analyzers temperature at 150 °C. High-purity nitrogen gas was used as collision-activated dissociation gas, and set at flow rate of 1.5 mL/min. The optimized dMRM parameters were summarized in Appendix A.

### 3.6. Method Validation

The developed GC-MS/MS method combined with SPE and MAD was validated in terms of limits of detection (LODs) and quantification (LOQs), linearity, precision, and accuracy. In this study, the use of five isotopically labeled ISs enabled alleviation of unfavorable results such as loss of steroids during sample pretreatment and variable detection sensitivity. The LODs and LOQs were defined as concentration levels over signal-to-noise (S/N) ratio at 3 and lowest concentrations satisfied with acceptable precision and accuracy, respectively. Calibration curves for 20 steroids were constructed with eight different concentration points. The determination coefficients (*r*^2^) were investigated over calibration ranges for all target analytes. The intra- and inter-day assay was employed to determine precision and accuracy of the developed analytical method with low, medium, and high concentrations in five replicates. All validation results demonstrated that the developed method is reliable and accurate for 20 endogenous steroids in human plasma samples. Overall validation data was summarized in Appendix A.

## 4. Conclusions

In this study, an SPE-MAD-GC-MS/MS-dMRM method was developed to profile 20 endogenous steroid hormones in human plasma. Under the optimized conditions respect to irradiation power and time, the MAD method could provide highly sensitive determination of target steroids. Two hundred microliter of plasma sample was sufficient to determine target analytes using C18-SPE cleanup. Moreover, dMRM method could yield high detection sensitivity of target steroids compared to traditional MRM method. The established method was successfully applied to human plasma with CSG (*n* = 20), IM (*n* = 13), or gastric cancer (*n* = 33). According to the profiling results for steroids, significant difference of E2 plasma levels between gastric disorder groups were identified. Moreover, by investigation of step-by-step metabolism state, it was demonstrated that metabolism pathways related to CYP11B1 and HSD17B were significantly altered between gastric disorder groups. Interestingly, E2 and HSD17B could be related to cancerous diseases. Therefore, this method is expected to be a useful method to investigate metabolic alterations of steroid metabolism pathways in human plasma and would be helpful to understand pathogenic mechanisms of gastric disorders and discover potential biomarkers for gastric disorders.

## Figures and Tables

**Figure 1 ijms-22-01872-f001:**
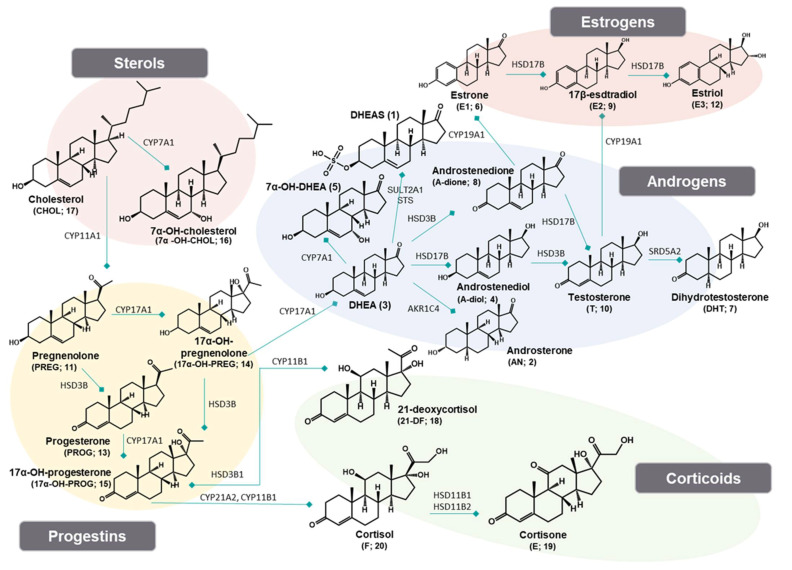
Metabolic pathways of 20 endogenous steroids.

**Figure 2 ijms-22-01872-f002:**
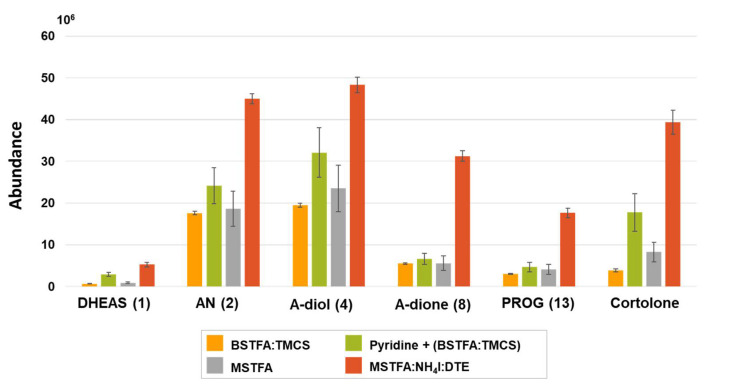
Comparison of yields of steroid derivatives for representative derivatization reagents (DHEAS: dehydroepiandrosterone sulfate; AN: androsterone; A-diol: androstenediol; A-dione: androstenedione; PROG: progesterone).

**Figure 3 ijms-22-01872-f003:**
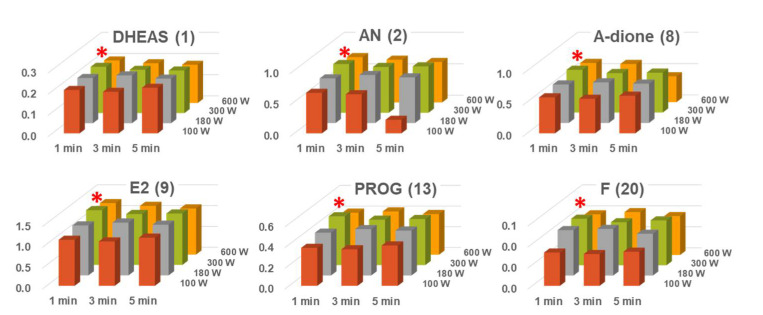
Influences of irradiation power and time in microwave-assisted derivatization (*: optimal derivatization conditions; DHEAS: dehydroepiandrosterone sulfate; AN: androsterone; A-dione: androstenedione; E2: 17β-estradiol; PROG: progesterone; F: cortisol).

**Figure 4 ijms-22-01872-f004:**
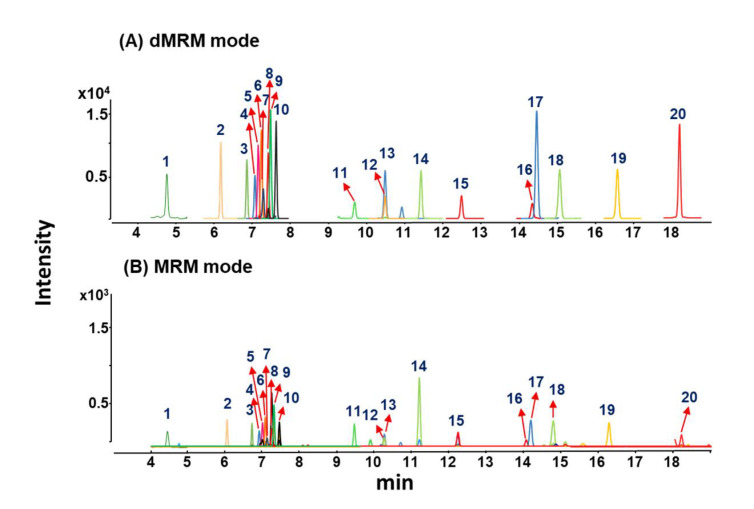
MRM chromatograms of 20 steroids using (**A**) a dMRM mode and (**B**) a traditional MRM mode. (The peak identified as follows: 1: DHEAS; 2: AN; 3: DHEA; 4: A-diol; 5: 7-OH-DHEA; 6: E1; 7: DHT; 8: A-dione; 9: E2; 10: T; 11: PREG; 12: E3; 13: PROG; 14: 17α-OH-PREG; 15: 17α-OH-PROG; 16: 7α-OH-CHOL; 17: CHOL; 18: 21-DF; 19: E; 20: F).

**Figure 5 ijms-22-01872-f005:**
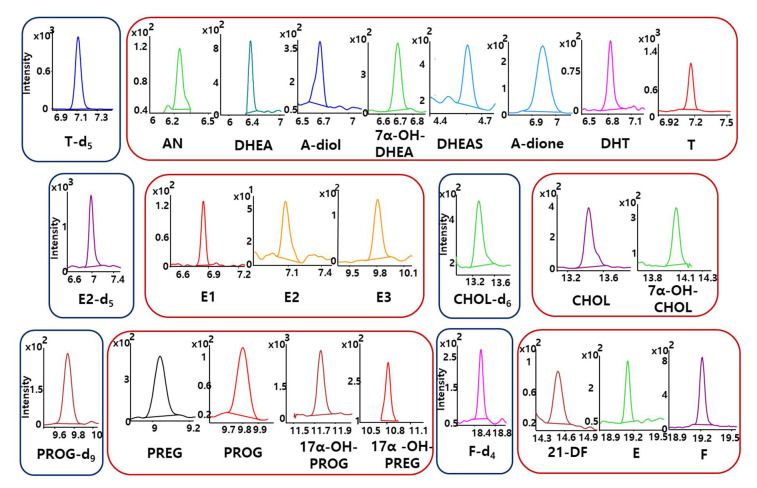
MRM chromatograms of 20 steroids and 5 isotopically labeled internal standards in human plasma sample. (AN: androsterone; DHEA: dehydroepiandrosterone; A-diol: androstenediol; 7α-OH-DHEA: 7α-hydroxydehydroepiandrosterone; DHEAS: dehydroepiandrosterone sulfate; A-dione: androstenedione; DHT: dihydrotestosterone; T: testosterone; E1: estrone; E2: 17β-estradiol; E3: estriol; CHOL: cholesterol; 7α-OH-CHOL: 7α-hydroxycholesterol; PREG: pregnenolone; PROG: progesterone; 17α-OH-PROG: 17α-hydroxyprogesterone; 17α-OH-PREG: 17α-hydroxypregnenolone; 21-DF: 21-deoxycortisol; E: cortisone; F: cortisol; T-d_5_: testosterone-2,2,4,6,6-d_5_; E2-d_5_: 17β-estradiol-2,4,16,16,17-d_5_; CHOL-d_6_: cholesterol-2,2,3,4,4,6-d_6_; PROG-d_9_: progesterone-2,2,4,6,6,17α,21,21,21-d_9_; F-d_4_: cortisol-9,11,12,12-d_4_).

**Figure 6 ijms-22-01872-f006:**
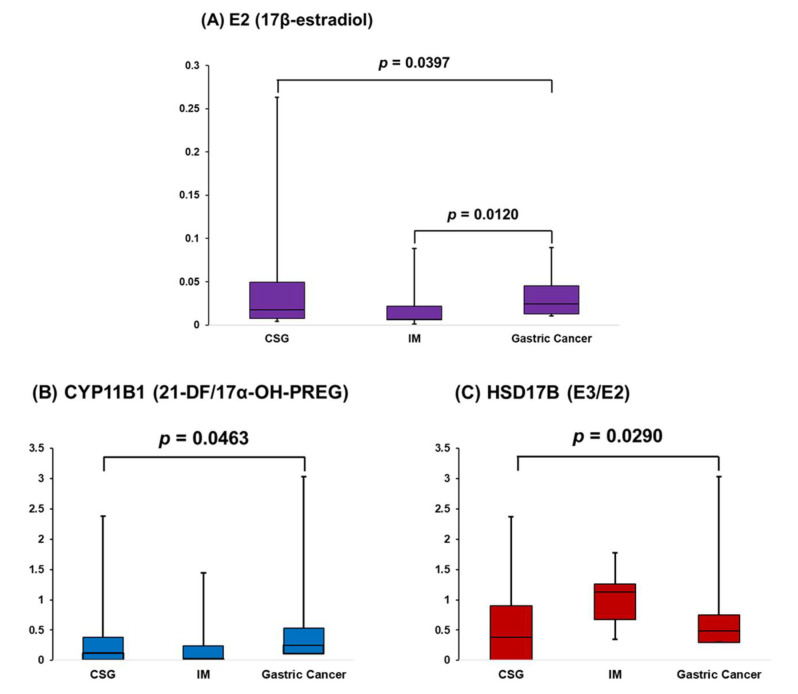
Metabolism alterations between CSG, IM, and gastric cancer. (**A**) E2 (17β-estradiol); (**B**) CYP11B1 (21-DF/17α-OH-PREG); (**C**) HSD17B (E3/E2).

**Table 1 ijms-22-01872-t001:** Concentrations and *p*-values of 20 steroids in human plasma samples.

Number	Name	CSG (ng/mL)	IM (ng/mL)	GC (ng/mL)	*p*-Values *
1	DHEAS	200.031 ± 228.170	186.637 ± 145.415	144.972 ± 167.46	0.2108
2	AN	0.800 ± 0.01	0.800 ± 0.006	0.803 ± 0.014	0.9960
3	DHEA	0.568 ± 0.019	0.565 ± 0.015	0.562 ± 0.012	0.4073
4	A-diol	0.535 ± 0.062	0.554 ± 0.103	0.587 ± 0.340	0.6514
5	7α-OH-DHEA	0.414 ± 0.045	0.542 ± 0.163	0.442 ± 0.067	0.9361
6	E1	0.937 ± 0.002	0.937 ± 0.001	0.937 ± 0.001	0.7711
7	DHT	0.224 ± 0.003	0.225 ± 0.002	0.224 ± 0.004	0.1146
8	A-dione	0.216 ± 0.014	0.214 ± 0.005	0.214 ± 0.006	0.2579
9	E2	1.533 ± 0.005	1.531 ± 0.002	1.532 ± 0.002	0.0397
10	T	0.312 ± 0.010	0.313 ± 0.008	0.317 ± 0.025	0.1522
11	PREG	1.688 ± 1.896	1.228 ± 0.671	1.034 ± 0.232	0.2030
12	E3	0.306 ± 0.00048	0.306 ± 0.00026	0.306 ± 0.0004	0.9436
13	PROG	1.677 ± 1.511	40.625 ± 125.020	1.628 ± 1.845	0.9954
14	17α-OH-PREG	0.348 ± 0.256	0.228 ± 0.053	0.269 ± 0.131	0.6539
15	17α-OH-PROG	2.344 ± 2.768	0.228 ± 0.053	1.284 ± 1.450	0.9679
16	7α-OH-CHOL	0.680 ± 0.492	0.591 ± 0.046	0.634 ± 0.348	0.3760
17	CHOL	0.088 ± 0.010	0.085 ± 0.008	0.082 ± 0.012	0.4055
18	21-DF	<LOQ	0.030 ± 0.001	0.031 ± 0.002	0.2637
19	E	35.925 ± 47.760	10.399 ± 9.193	42.649 ± 58.608	0.1724
20	F	29.913 ± 50.956	13.460 ± 11.376	34.821 ± 41.423	0.2630

* The *p*-values were determined with a Kruskal–Wallis test. (DHEAS: dehydroepiandrosterone sulfate; AN: androsterone; DHEA: dehydroepiandrosterone; A-diol: androstenediol; 7α-OH-DHEA: 7α-hydroxydehydroepiandrosterone; E1: estrone; DHT: dihydrotestosterone; A-dione: androstenedione; E2: 17β-estradiol; T: testosterone; PREG: pregnenolone; E3: estriol; PROG: progesterone; 17α-OH-PREG: 17α-hydroxypregnenolone; 17α-OH-PROG: 17α-hydroxyprogesterone; 7α-OH-CHOL: 7α-hydroxycholesterol; CHOL: cholesterol; 21-DF: 21-deoxycortisol; E: cortisone; F: cortisol).

**Table 2 ijms-22-01872-t002:** *p*-values of concentration ratios of plasma steroids from patients with CSG, IM, and gastric cancer.

Enzymes ^1^	*p*-Values ^2^	*p*-Values ^3^		
		CSG:IM	CSG:Gastric Cancer	IM:Gastric Cancer
CYP7A1 (7α-OH-CHOL/CHOL)	0.9611	0.7737	0.9627	0.8322
CYP11A1 (PREG/CHOL)	0.6121	0.3109	0.0596	0.583
HSD3B1 (PROG/PREG)	0.3408	0.2037	0.2042	0.8167
CYP17A1 (17α-OH-PROG/PROG)	0.3094	0.2937	0.9204	0.1100
CYP17A1 (17α-OH-PREG/PREG)	0.8078	0.8394	0.5061	0.8357
CYP11B1, CYP21A2 (F/17α-OH-PROG)	0.0854	0.6451	0.1457	0.0370
HSD11B1,2 (E/F)	0.1599	0.0936	0.4948	0.1074
CYP11B1 (21-DF/17α-OH-PREG)	0.0463	0.606	0.0546	0.0393
CYP17A1 (DHEA/17α-OH-PREG)	0.7316	0.5730	0.7642	0.4568
CYP7A1 (7α-OH-DHEA/DHEA)	0.8481	0.8250	0.8051	0.5179
SULT2A1, STS (DHEAS/DHEA)	0.6698	0.7575	0.3868	0.6873
AKR1C4 (AN/DHEA)	0.4075	0.2653	0.9348	0.2090
HSD17B (A-diol/DHEA)	0.7932	0.5730	0.5789	0.8548
HSD3B (T/A-diol)	0.4422	0.2936	0.3028	0.6170
SRD5A2 (DHT/T)	0.2404	0.8683	0.2366	0.1213
HSD3B (A-dione/DHEA)	0.7530	0.4282	0.5695	0.9903
CYP19A1 (E1/A-dione)	0.6811	0.6584	0.7068	0.3864
CYP19A1 (E2/T)	0.0871	0.0235	0.4608	0.1128
HSD17B (E3/E2)	0.0290	0.0213	0.5300	0.0168

1: Abbreviations of enzymes: CYP, cytochrome P450; HSD, hydroxysteroid dehydrogenase; SULT, sulfotransferase; AKR, aldo-keto reductase; SRD, steroid-5α-reductase; 2: The *p*-values were determined with a Kruskal–Wallis test; 3: The *p*-values were determined with a Mann–Whitney U test.

## Data Availability

Data available in a publicly accessible repository.

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
