# Peer review of "Profiling of Steroid Metabolic Pathways in Human Plasma by GC-MS/MS Combined with Microwave-Assisted Derivatization for Diagnosis of Gastric Disorders"

_ijms, 2021, doi:10.3390/ijms22041872_

Round 1

Reviewer 1 Report

The manuscript represents an interesting comprehensive approach to determine a long series of endogenous steroids in human plasma to profile metabolism pathways as a preliminary prognosis of selected diseases. The advanced analytical methodology was applied combined with solid-phase extraction and chemical derivatization. As suggested correctly by the authors, the method developed might be useful in identifying potential biomarkers for the development of diseases under study. The extensive screening of various classes of steroids identified a single estrogen and the two metabolic pathways possibly indicative of gastric disorders. 

There are no major concerns and the manuscript deserves publication after minor corrections and moderate language improvements.

Minor comments:

- names od steroids, abbreviations, and numbers should be closely correlated in figures and Tables 

Fig. 1 - numbers for steroids  shown in Fig. 4, should also be included in Fig. 1, for easy correlation of compounds on chromatograms with their structures

Fig. 2. and 3. - abbreviations for steroids should be added to the figure legend, for the convenience of the reader.

Table 1 - introduce steroid numbers

Table 3 - move the method validation results to SI

- more references from the last three years should be cited to prove the subject of the manuscript is of a current research interest

Author Response

Dear reviewer I:

I am truly grateful to your critical comments and valuable suggestions. We have revised the manuscript based on your comments and suggestion. I hope the revised manuscript will be acceptable for publication in International Journal of Molecular Sciences.

Thank you very much for your valuable comments to our paper.

Yours sincerely,

Prof. Jongki Hong

College of Pharmacy, Kyung Hee University

Our alterations as a result of the reviewer’s comments are:

==============================================================

Reviewer #1

The manuscript represents an interesting comprehensive approach to determine a long series of endogenous steroids in human plasma to profile metabolism pathways as a preliminary prognosis of selected diseases. The advanced analytical methodology was applied combined with solid-phase extraction and chemical derivatization. As suggested correctly by the authors, the method developed might be useful in identifying potential biomarkers for the development of diseases under study. The extensive screening of various classes of steroids identified a single estrogen and the two metabolic pathways possibly indicative of gastric disorders. 

There are no major concerns and the manuscript deserves publication after minor corrections and moderate language improvements.

→ Before submission, English grammar of manuscript was carefully proofread by the professional editing company. If possible, could you kindly inform me grammatical or typesetting mistakes?

Minor comments:

- names od steroids, abbreviations, and numbers should be closely correlated in figures and Tables 

Fig. 1 - numbers for steroids  shown in Fig. 4, should also be included in Fig. 1, for easy correlation of compounds on chromatograms with their structures

Table 1 - introduce steroid numbers

→ As above reviewer’s suggestions, we double-checked names, abbreviations, and peak number of steroids in Figures and Tables. We described peak number of steroids in Tables and Figures to help the readability of manuscript.

Fig. 2. and 3. - abbreviations for steroids should be added to the figure legend, for the convenience of the reader.

→ As reviewer pointed out, we revised figure captions to include abbreviations for steroids.

Table 3 - move the method validation results to SI

→ As reviewer suggested, we moved Table 3 into supplementary data.

- more references from the last three years should be cited to prove the subject of the manuscript is of a current research interest

→ As reviewer suggested, we cited several recent literatures published in 2019 and 2020 on steroid profiling [such as J. Pharm. Biomed. Anal. 170 (2019) 161-168, J. Pharm. Biomed. Anal. 175 (2019) 112756, and J. Steroid Biochem. Mol. Biol. 198 (2020) 105615].

Reviewer 2 Report

The submitted manuscript contains a large amount of measured data, which are well presented.

Nevertheless, it would be appropriate to edit some images and comment on them appropriately in the text.

Figure 2 does not show what the units are on the x-axis. How many times were derivatization experiments performed? Why was this group of substances chosen for the derivatization test?

It would be appropriate to show in Figure 2 the variance of the individual derivatization experiments' values.

From the results shown in Figure 3, it is not possible to determine without a statistical background whether microwave derivatization had any effect. What does a star symbol mean?

What is the difference between MRM and dMRM mode?

I did not find any data from the SPE extraction testing procedure. There is only a reference in the manuscript without a summary of the cited study's data.

Author Response

Dear reviewer II:

I am truly grateful to your critical comments and valuable suggestions. We have revised the manuscript based on your comments and suggestion. I hope the revised manuscript will be acceptable for publication in International Journal of Molecular Sciences.

Thank you very much for your valuable comments to our paper.

Yours sincerely,

Prof. Jongki Hong

College of Pharmacy, Kyung Hee University

Our alterations as a result of the reviewer’s comments are:

==============================================================

Reviewer #2:

The submitted manuscript contains a large amount of measured data, which are well presented.

Nevertheless, it would be appropriate to edit some images and comment on them appropriately in the text.

Figure 2 does not show what the units are on the x-axis. How many times were derivatization experiments performed? Why was this group of substances chosen for the derivatization test?

→ In Figure 2, the unit of the x-axis on graph was none, but names of compounds and reagents. Furthermore, we rearranged the x-axis according to steroid number. And the unit of y-axis was peak abundance for steroid derivatives. In our test, overall experiments were performed in triplicate and several steroids were randomly selected according to their structural characteristics.

It would be appropriate to show in Figure 2 the variance of the individual derivatization experiments' values.

→ As reviewer suggested, error bar was added in Figure 2.

From the results shown in Figure 3, it is not possible to determine without a statistical background whether microwave derivatization had any effect. What does a star symbol mean?

→ In Figure 3, it seemed that multiple microwave-assisted derivatization conditions presented similar derivatization yields. However, star marked conditions showed slightly higher and reproducible compared to other conditions. Therefore, considering both derivatization time, yields, and reproducibility, we selected star marked conditions as optimized microwave derivatization conditions. In conclusion, star marked conditions were optimal conditions.

What is the difference between MRM and dMRM mode?

→ Dynamic MRM methods (Agilent) also called as time scheduled MRM methods (AB Sciex). In this study, we called dMRM mode, since we used an Agilent GC-MS/MS instrument. dMRM methods can utilize more flexible dwell time and retention time windows according to compound numbers between retention time windows, compared to traditional MRM methods. In section 2.2. GC-MS/MS-dMRM, we already provided concise description of dMRM method. Detailed information about dMRM can be obtained from Agilent online website.

I did not find any data from the SPE extraction testing procedure. There is only a reference in the manuscript without a summary of the cited study's data.

→ In this study, we utilized the SPE protocol previously reported in PLoS One 7 (2012) e32496. This previous study reported that optimized SPE protocol provided extraction efficiency ranged from 87% to 101%. Since we reproduced these results using slightly modified SPE protocol, SPE recovery results were not presented in this manuscript. As reviewer pointed out, we put add description of the cited study’s data in revised manuscript.